# Secure FLOATING - Scalable Federated Learning Framework for Real-time Trust in Mobility Data using Secure Multi-Party Computation and Blockchain

## Abstract

The safety of Connected and Autonomous Vehicles (CAVs), Micro-mobility devices (e-scooter, e-bikes) and smartphone users rely on trusting the trajectory data they generate for navigation around each other. There is a need for real-time verification of mobility data from these devices without compromising privacy as malicious data used for navigation could be deadly, specially for vulnerable road users. In this paper, we propose Secure-FLOATING, a scalable framework leveraging federated learning and blockchain for nearby nodes to coordinate and learn to trust mobility data from nearby devices and store this information via consensus on a tamper-proof distributed ledger. We employ lightweight Secure Multi-party computation (SMPC) with reduced messages exchanges to preserve privacy of the users and ensure data validation in real-time. Secure-FLOATING is evaluated using realistic trajectories for up to $8,000$ nodes (vehicles, micro-mobility devices and pedestrians) in New York City, and it shows to achieve lower delays and overhead, thereby accurately validating each others' mobility data in a scalable manner, with up to 75% successful endorsement for as high as 50% attacker penetration.

## 1 Introduction

Connected and Autonomous Vehicles (CAVs) and micro-mobility devices (e-scooters, e-bikes, etc.) are poised to transform transportation, promising increased safety, efficiency, and accessibility. We envision a scenario where CAVs and other road users specially Vulnerable Road Users (VRUs) such as pedestrians with smartphones and e-bike/ e-scooters riders share their intended trajectories for the next few seconds or minutes with nearby road users and coordinate navigation around each other in real time. Imagine a signal-less intersection with CAVs and other road users safely and seamlessly navigate around each other by sharing sensory and future trajectory data with each other prior to crossing the intersection, thereby communicating intention along the exact speed, location and other information needed for intersection crossing. However, nodes will hesitate, unable to ascertain the intentions of their counterparts, or worse, where malicious actors could exploit system vulnerabilities. In addition to faulty sensors, false data generation could mislead the users relying on it for their real-time navigation decisions around CAVs, and therefore could lead to deadly consequences. It is therefore vital need for a robust, reliable, and trustworthy decision-making framework within the CAV ecosystem.

Current trust models, often centralized in nature are inadequately designed for such networks. Centralizing vast amounts of sensitive mobility data creates a prime target for attacks and privacy breaches, while relying on a single point of authority introduces a dangerous single point of failure. Additionally, the opaque nature of centralized models leaves room for manipulation, potentially compromising the integrity of the entire network. Stronger anonymization and encryption techniques can achieve strong privacy but reduce the usefulness of the data in time and safety-critical scenarios due to their complexity and time consuming

computation involved resulting in high computation and communication overhead. Moreover, such approaches focuses solely on privacy preservation and ignore the data trustworthiness challenge with no method to detect or mitigate impact of faulty or compromised sensors generating misleading information resulting in dissemination of incorrect or manipulated information.

We introduce Secure-FLOATING: **F**ederated **L**earning for **O**ptimized **A**utomated **T**rust **ING**, a blockchain-based platform as a paradigm shift in trust establishment for real-time interactions regarding navigation between CAVs and other road users to collaboratively learn representation of each others mobility locally without revealing data. It is a novel synergistic combination of Verifiable Federated Learning (VFL), Secure Multi-Party Computation (SMPC) as a scalable and privacy-aware technique targeted at real-time validation of mobility data from CAVs and other devices through a blockchain based consensus, ensuring safety of road users. However, doing so in real-time is challenging due to the increasing computation and communication overhead of SMPC, the consensus process along the frequency of model aggregation rounds in federated learning, in addition to the complexity of underlying prediction models. Secure-FLOATING is a lightweight solution to minimize this overhead from multiple aspects. First, the choice of a simple addition-based function for SMPC, although Secure-FLOATING is not limited to addition-based function and a complex function could be used based on the application requirements. Second, we suggest lightweight models for the predictions of trajectories and show that such models performs comparable with relatively complex ones, without the added computation overhead measured in FLOPs. Third, our theoretical analysis show that Secure-FLOATING is scalable with linear increase in overhead with the increase in network size (number of nodes), while ensuring privacy guarantees. and finally, we are the first ones to set the basis for reducing message exchange rounds for the SMPC, consensus process, and model updates in federated learning to further reduce communication overhead.

At the heart of Secure-FLOATING lies VFL, a novel approach that enables collaborative learning of a trust model across vehicles without the need for sharing raw data. This preserves privacy while ensuring the integrity of each node's contribution through cryptographic verification mechanisms. To securely aggregate model updates from multiple nodes without revealing sensitive information, we employ a lightweight MPC protocol to guarantee the confidentiality of individual contributions while maintaining the accuracy of the global model. Secure-FLOATING employs a consensus mechanism, allowing nodes to collaboratively validate each other's trajectory data for real-time navigation decisions. We leverage zero knowledge proof as "trust-but-verify" mechanism to filter out noisy or malicious data, thereby significantly enhancing the reliability of the trust model. We provide rigorous theoretical guarantees, proving that the communication overhead of this mechanism scales linearly with the number of vehicles, making it eminently suitable for large-scale deployments. It also leverages the decentralized nature of blockchain or the InterPlanetary File System (IPFS) (2) to store the global trust model and validation results. This ensures transparency, immutability, and auditability of the system, further bolstering its trustworthiness.

The paper organization is such that the following section discusses the related work and positions the novelty of our contribution. Section 3 describes the proposed Secure-FLOATING platform, followed by a comprehensive theoretical analysis in Section 4. Section 5 evaluates the platform, Section 6 discusses ethical considerations and limitations with future insight. Section 7 concludes the paper.

## 2 Related Work

Trust establishment in CAV networks is a critical area of research, with various models proposed to address the challenges of safety, cooperation, and reliability. Centralized trust models, such as those based on reputation scores (e.g., Dhurandher et al. (8), Liu et al. (15)) or data-driven approaches (e.g., using deep learning by Yao et al. (26) and game-theoretic approaches by Asvadi et al. (1)), have been explored. However, they suffer from inherent limitations. Centralized data aggregation raises privacy concerns, as sensitive vehicle data is transmitted to and stored in a central location (19). Additionally, these models are vulnerable to single points of failure and can be susceptible to manipulation by malicious actors (18).

Decentralized trust models have emerged as a promising alternative, empowering individual vehicles to collaboratively build trust without relying on a central authority. Blockchain-based solutions have been proposed to leverage the decentralized and immutable nature of the technology for trust management in CAVs (20; 25). However, these approaches often focus on transaction verification and data integrity, without adequately addressing the privacy concerns associated with sharing raw data on the blockchain (3).

Federated learning (FL) has also been explored for trust model building in CAVs (16; 23). FL enables collaborative learning while keeping data local to each vehicle, enhancing privacy. However, existing FL-based trust models often lack robustness against malicious participants and do not provide formal guarantees on the security of the aggregation process (6). VFL extends traditional FL by incorporating mechanisms to verify the correctness of model updates contributed by individual clients (14). Zero-knowledge proofs (ZKPs) have been widely used in VFL to enable verification without revealing the actual model updates (5; 24). However, existing VFL approaches primarily focus on general machine learning tasks and may not be optimized for the specific challenges of trust model building in CAVs, such as handling heterogeneous data distributions, ensuring robustness against specific attacks (e.g., Sybil attacks), and incorporating real-time validation of data like inter-vehicle distances.

SMPC has been utilized in various domains for privacy-preserving data aggregation and analysis (7). In the context of CAVs, SMPC has been proposed for secure intersection management (17) and cooperative perception (13), while (9) proposed secret sharing in federated learning, the threat model is different where the authors simplify assumption of considering an "honest but curious" node instead of a malicious one. We target real-time validation of malicious data shared among vehicles leveraging blockchain or distributed ledger for tasks that are time sensitive, such as navigating safely around each other. Nevertheless, traditional SMPC based approaches suffer from their complexity and overhead making them unsuitable for real-time safety-critical applications where split-second decision- making is paramount on resource constrained CAVs, thus, facing a trade-off between achieving privacy vs. utility at scale.

We propose a novel framework with lightweight addition-based SMPC integrated with VFL as a secure consensus-based trajectory validation mechanism to achieve decentralized, privacy-preserving, and robust trust establishment. This is critical for the safety of road users, where wrong trajectory estimation could have fatal consequences, specially on VRUs. One of the biggest challenge currently to achieve real-time requirements is to reduce the time in computing the functions used in SMPC. The use of addition-based SMPC is to leverage and show a simple (linear) function for nodes to compute in order to comply to real-time requirements of the application towards a lightweight validation. However, Secure-FLOATING is not limited to an additive function and a more complicated function could be used with different time requirements depending on the complexity of function, where the time it takes to run the function is proportional to its complexity. There exist no paper in the literature to address real-time CAVs data validation, critical for time-sensitive safety applications for CAV. Secure-FLOATING is the first trust model that goes beyond traditional offline cyber attack detection solutions in CAVs with simple assumptions and threat models and instead targets real-time malicious data validation in a scalable manner, evaluated on realistic trajectory data involving $8,000$ connected vehicles in New York City.

## 3 Secure FLOATING Framework

### 3.1 System Overview

The system model considers a set $V = \{v_1, v_2, ... v_n\}$ representing nodes (i.e. CAVs, micro-mobility sources, pedestrians with smartphones, etc. ) in the wireless communication range of each other. The communication range can differ depending on the underlying wireless communication technology. For ease of implementation, the communication model between nodes we assume considers but not limited to wireless short range communication technologies such as the 2.4/5GHz IEEE 802.11 standard or 5.9 GHz Cellular Vehicle-to-Everything (C-V2X) recently adapted by US DOT for V2X communication.

We define a node $v$ trajectory as, $C_v(l, t) = <(l_v^1, t_v^1), (l_v^2, t_v^2), ...(l_v^m, t_v^m)>$ with sequence of locations, $l_v^1, l_v^2, ..$ at times $t_v^1, t_v^2, ..,$ respectively. The location information composed of the geographical latitude and longitude $l_v = (lat_v, lon_v)$ as GPS coordinates. We assume nodes are synchronized to a common (GPS) clock with the time $T = \{t_0, t_1, t_2, ...\}$ as consecutive time instants $t_0, t_1, t_2...$ and so on. We consider, at time $t_0$, a set of nearby nodes, $v_1, v_2, v_3$, exchange sequence of trajectories $C_{v_1}, C_{v_2}, C_{v_3}$, representing the location points they intend be at in the next few time instants $t_1, t_2..$ to coordinate safe navigation around each other.

## THREAT MODEL

The threat model we consider is compromised node $v$'s trajectory data $C_v(l, t)$, resulting in either the node generating fake or malicious trajectories that when shared with nearby vehicles for real-time navigation decisions could lead to deadly consequences. To address this, we define a blockchain based zero trust protocol for nearby nodes to verify trajectory data shared by nearby nodes and endorse each others trajectories at regular time intervals on a distributed ledger, where the frequency at which they are endorsed depending on the application and resource consumption requirements, however, we consider it to be less frequent in order to conserve the computing and communication resources on the nodes.

The blockchain model we consider is a permissioned blockchain where nodes are authenticated prior joining the network using state of the art public/private key based authentication or a similar encryption mechanism, however, developing such system is out of the scope of the paper. We employ the Inter Planetary File System (IPFS) (2), a distributed storage system for the nodes to store the endorsement information on an immutable distributed ledger. Interested readers can refer to (10) (11) for Reinforcement Learning (RL) based optimized frequency of endorsements and updates on the distributed ledger or blockchain.

## 3.2 VERIFIABLE FEDERATED LEARNING

We employ federated learning for the verification of trajectories enabling each node to locally run lightweight models leveraging a local representation based on its own data to predict corresponding future trajectories of nearby vehicles (i.e. at times $t_1, t_2... > t_0$) and endorse the node as trustworthy if the predicted trajectories match the shared one within an acceptable margin or error.

The local prediction models at a node $v_i$ train to minimize the loss function as:

$$\theta_{v_i} \leftarrow \theta_{v_i} - \eta \nabla_{\theta_{v_i}} L(\theta_{v_i}, D_{v_i})$$

where, $\theta_{v_i}$ represents local model parameters for node $v_i$, $\eta$ is the learning rate and $\nabla_{\theta_{v_i}} L(\theta_{v_i}, D_{v_i})$ represent the gradient of the loss function $L$ for node $v_i$, computed using its local dataset $D_{v_i}$.

The global model aggregation (FedAvg or similar) of the locally trained models from different node is regularly updated on the blockchain as:

$$\theta_{global} \leftarrow \frac{\sum_{v_i}(n_{v_i}\theta_{v_i})}{\sum_{v_i} n_{v_i}}$$

where, $\theta_{global}$ represents the global model parameters and $n_{v_i}$ are the number of samples from node $v_i$.

We present a Verifiable Federated Learning (VFL) approach using SMPC to avoid the model updates (i.e. gradients) from being manipulated by the nodes. Each node splits the local model updates $\theta_{v_i}$ into shares $s_{v_{i_1}}, s_{v_{i_2}}... s_{v_{i_k}}$, for $k$ nodes. The shares are generates such that $\theta_{v_i} = s_{v_{i_1}} + s_{v_{i_2}}... + s_{v_{i_k}}$, and any subset of less than $k$ shares reveals no information about $\theta_{v_i}$. The node $v_i$ sends share $s_{ij}$ to neighboring node $v_j$. Each node $v_j$ sums up the share it received from all other nodes to get the aggregated model update $\theta_{v_j} = \sum_{v_i} s_{v_{ij}}$. Finally each node $v_j$ sends its aggregated share $\theta_{v_j}$ to the blockchain to obtain the global model update:

$$\theta_{global} = \sum_{v_j} \theta_{v_j}$$

We consider all nodes participating in the SMPC based VFL approach employ the same model to ensure consistency in representing model updates, however the exact prediction model the nodes agree to use could vary since an application could require different methods of predicting trajectories of nearby nodes. For instance, a vehicle can use GPS and/or other sensors data (LiDARs, cameras, etc.) to observe neighboring node and predict its next coordinates. Similarly, a node could use part of the currently shared trajectories or the history of the previously shared trajectories to predict what the future trajectories would be for a given node in its surroundings. We leave the choice of the prediction algorithm open, as the success of Secure-FLOATING is independent of the underlying prediction algorithm, though we prefer simple models compared to resource intensive ones.

### Addition-based SMPC

To better understand the above SMPC protocol for model updates sharing, we describe a toy problem for a simple addition based secure multi-party computation function below. Consider three nodes, $v_1, v_2, v_3$, with reference distances between $v_1$ $d(v_1, v_1) = 0$, $d(v_1, v_2) = 5.5$ and $d(v_1, v_3) = 4.5$, where $v_1$ is the consensus initialization node. The addition-based encryption in node $v_1$ splits 0 randomly into $-1, 1$ and 0. It keeps $-1$ and sends 1 to $v_2$ and 0 to $v_3$ respectively. Similarly $v_2$ splits its distance 5.5 randomly into 2,3 and 0.5. It sends 2 to $v_1$ keeps 3 at $v_2$ and sends 0.5 to $v_3$. Node $v_3$ randomly generates 5, $-1$ and 0.5, sends 5 to $v_1$, $-1$ to $v_2$ and keeps 0.5. The node $v_1$ computes the sum of $-1, 2$, and 5 as 6, $v_2$ computes the sum of $1, 3$, and $-1$ as 2, and $v_3$ computes the sum of $0, 0.5, 0.5$ as 1. The three nodes exchange each other's computed sum resulting in a total sum of $(6 + 2 + 1 = 10)$, with a function to compute mean value of 3.33. The nodes then exchange the mean value among themselves, where a match in values at each node would serve as validation of the transaction. In case any of these value differ at the nodes, the consensus process is considered void and the corresponding nodes are untrustworthy.

The above toy problem uses an addition-based function, however Secure-FLOATING will work with any function computed and matched among peers, and therefore we leave the choice of the function open to the specific application/ use case where simple functions are preferred to reduce complexity.

### 3.3 Endorsement on Distributed Ledger

The workflow for the endorsement process via consensus is summarized as follows:

*Step 1.* Each node exchanges future trajectory $C_v$ with neighboring nodes in its wireless communication range.

*Step 2.* It employs a lightweight federated learning model to predict future trajectories for each of its neighboring nodes by levering either the on-board sensors, or the trajectories shared by the neighbor using the global model gradients ($\theta_{global}$) retrieved from the distributed ledger.

*Step 3.* It compares the predicted trajectories with the ones shared by the neighbor $v_j$ between two time instants or the interval $\Delta t = [t_0, t_l]$, and if the difference is within an acceptable threshold $\omega$, it endorses the truthfulness of the corresponding neighboring node.

$$\frac{1}{|\Delta t|} \sum_{\Delta t \in [t_0, t_l]} C_{v_j}(l, t)[shared] - C_{v_j}(l, t)[predicted] \leq \omega$$

*Step 4.* It splits the model updates ($\theta_{v_i}$) using SMPC and exchanges shares of it with neighbors running the same federated learning model.

*Step 5.* It combines the shares of the model updates received from neighboring nodes and updates the distributed ledger (IPFS) with the global model along the node IDs of the nodes for which it endorsed the respective trajectories.

*Step 6.* To reach consensus, the aggregated model updates from nodes for whom the majority of neighbors (51%) endorsed the trajectories is updated as the global model on the distributed ledger along storing the respective node IDs.

## 4 THEORETICAL ANALYSIS

**Theorem 4.1. (Privacy Guarantee)** *Let ($\epsilon > 0$) be a privacy budget and ($\delta > 0$). Under our proposed SMPC aggregation protocol using addition-based secret sharing, the aggregated model update satisfies (($\epsilon, \delta$))-differential privacy.*

*Proof.* (Appendix A.1)

**Theorem 4.2. (Scalability Analysis)** *Each node (i) shares trajectory with ($n-1$) nodes, (ii) exchanges secret shares with ($n-1$), and subsequently updates the distributed ledger with a single message, therefore, the total communication overhead for n nodes in the network is represented by the function $f(n) = (n-1) + (n-1) + 1 = 2n-1$, which exhibits a linear growth with respect to n.*

*Proof.* (By induction)

Base Case ($n = 1$): The function holds true for the initial value: ($f(1) = 3(1) - 2 = 1$).

**Inductive Hypothesis:** Assume the function holds true for an arbitrary positive integer k, i.e., ($f(k) = 3k - 2$).

**Inductive Step:** (n = k + 1):To prove ($f(k+1) = 3(k+1) - 2$), by definition, we know ($f(k+1) = 3(k+1) - 2 = 3k + 1$). Using the hypothesis, we can express this as: ($f(k+1) = (3k-2) + 3 = f(k) + 3$) This shows that the output increases by a constant amount (3) for each unit increase in the input.

## 5 EXPERIMENTAL EVALUATION

### DATASET

To evaluate the effectiveness and practicality of the proposed Secure-FLOATING platform, we used realistic trajectories generated from real nodes in the New York City connected vehicles pilot project (21) comprising of around $8,000$ vehicles as our training data. An open-source microscopic traffic simulation software, SUMO (12), and its internal extension API, Traffic Control Interface (TraCI) (22) are used to generate trajectories. SUMO can simulate intricate and realistic vehicular driving behavior as well as traffic performance, while TraCI enables online interaction with the primary simulation by connecting external applications to SUMO via sockets. The studied area encompasses a 1.6-mile stretch on Flatbush Avenue between Tillary Street and Army Plaza, recognized as one of the US DOT Connected Vehicle Pilot Program test sites located in Brooklyn, New York. The simulation parameters, such as acceleration rate, minimum gap, reaction time, distribution of different types of vehicles, and spatial and temporal features of local traffic flow, are regarded as random variables. They are calibrated as probability distributions founded on real trajectory data extracted from drone and CCTV camera videos, as well as actual local traffic data.

We considered the area between Fulton Street and DeKalb Avenue to analyze high traffic volume and capture significant urban driving behavior. The simulation time is divided into 72 segments, each consisting of 100-second intervals for a total of two hours duration. Mobility data for the heterogeneous set of nodes (comprising passenger cars, trucks, buses, e-bikes) are generated for the selected road section and time duration.

The trajectory data from SUMO is fed into an open source network simulator (NS3) to implement the wireless network among nodes, where different network configurations are defined to account for the number of neighbors in each other's wireless communication range based on their coordinates. We consider the communication range of 100 meters to define the neighbors for a node, i.e. nodes with coordinates that are within 100 meters are considered neighbors, however we varied the range from 50-300 meters to understand potential impacts and uncertainty from the wireless communication medium. The temporal neighbors are defined as the nodes that are within this communication range for at least 5 seconds. The rationale for this is to ensure nodes stay in each others communication range for a sufficient amount of time to complete the endorsement process.

BASELINE

Secure-FLOATING is not limited to a particular model and different machine learning models can be used for trajectory prediction. Although we recommend using lightweight models for their low computational complexity and lower delays, we compared an array of models, considering simple models such as vanilla RNN, LSTM, GRU, as well as complex models such as the Transformer (Informer (27)) and image based Object Detection with Assistance (ODA) models such as YOLO (4) for nodes to predict trajectories of nearby nodes to evaluate a variety of techniques for trajectory prediction, where nodes run each model with and without SMPC based VFL to highlight its benefits.

For the network overhead of message exchanges between node, we configured message sizes of 1kb and 100kb to be exchanged between nodes at different frequencies, notably, every 1 second, 10 seconds and 1 minutes respectively. Secure-FLOATING can also adapt RL-based adaptive approach (11) to the message exchanges for consensus, however, to demonstrate the low complexity, we are using fixed message sizes and frequency of exchanges between nodes endorsing each other's trajectories. For a similar reason, we considered four network configuration, neighborhoods of 5, 10, 30 and 50 nodes endorsing each others trajectories as we believe increasing the number of nodes endorsing each other will further increase the complexity.

METRICS

The key metrics we consider to evaluate all models in Secure-FLOATING are Mean Absolute Error, training time, accuracy in correct neighbors trajectory prediction, Floating Operations Per Second (FLOPs) and CPU utilization. The metric used for scalability analysis is the overhead computed with the increase in the network size, with varying message size (1kb,100kb) and exchange frequency (1s,10s,1min) as described above.

All data preparation and model runs are conducted on an Intel(R) Core(TM) i7-7700HQ CPU @ 2.80GHz Processor with 16 GB of installed RAM. The training and test data as well as models are optimized to train on a single NVIDIA 1670 GPU with 8 GB of VRAM made possible with PyTorch's CUDA toolkit version 11.8. All model runs are conducted with a Windows 10 operating system utilizing Python version 3.6. The machine learning framework used is PyTorch 1.13.1 with CUDA version 11.6.

RESULTS

| Model | MAE | Accuracy |
|---|---|---|
| LSTM | 6.3 | 93.91 |
| RNN | 7.2 | 92.29 |
| GRU | 6.1 | 95.37 |
| Transformer | 0.7 | 99.53 |
| ODA | 6.2 | 99.75 |

Table 1: Mean Absolute Error

| Model | Sequence Length | Params | FLOPs | CPU Util. % |
|---|---|---|---|---|
| LSTM | 480 | 335.49K | 43.34 | 57.7 |
| RNN | 480 | 83.97K | 10.78 | 6.4 |
| GRU | 480 | 251.65K | 32.54 | 50.8 |
| Informer | 96 | 32.3M | 59.12 | 92.2 |
| ODA | 480 | 7.5M | 49.40 | 78.79 |

Table 2: Model parameters

Table 1 show the MAE results and accuracy of each model for the trajectory prediction. By comparing the predicted trajectories with the ground truth data, we observed consistently

| Model | n=1 | n=5 | n=10 | n=30 | n=50 | n=100 |
|---|---|---|---|---|---|---|
| VFL-LSTM | 12.32 | 12.43 | 13.38 | 15.45 | 16.58 | 17.01 |
| LSTM | 7.03 | 31.12 | 66.34 | 202.56 | 340.42 | 699.87 |
| VFL-RNN | 11.87 | 12.01 | 12.79 | 13.93 | 15.45 | 16.38 |
| RNN | 7.06 | 38.92 | 64.25 | 197.10 | 336.08 | 712.99 |
| VFL-GRU | 11.65 | 12.27 | 13.03 | 14.45 | 15.01 | 15.66 |
| GRU | 7.09 | 31.20 | 62.90 | 195.26 | 344.69 | 687.38 |
| VFL-Informer | 25.38 | 26.83 | 30.03 | 32.88 | 35.54 | 39.77 |
| Informer | 14.10 | 90.90 | 191.50 | 583.20 | 971.40 | 1944.50 |

Table 3: Training time: ML versus VFL, 1-100 nodes

| MAE | 1 | 5 | 10 | 30 | 50 | 100 |
|---|---|---|---|---|---|---|
| VFL | 6.43 | 6.57 | 6.73 | 6.19 | 6.34 | 6.69 |
| Vanilla ML | 6.24 | 13.63 | 20.88 | 13.4 | 26.96 | 21.71 |

Table 4: MAE: ML versus VFL, 1-100 nodes

low MAE values across various models. This indicates that our lightweight models are fully capable of generating highly accurate predictions, thus instilling confidence in their reliability for node trajectory prediction tasks.

Moreover, we delved into the computational efficiency of Secure-FLOATING by comparing different models and examine the floating point operations per second (FLOPs) metric (Table 2). FLOPs serve as a measure of the number of floating-point operations required to perform the prediction task. Our findings revealed that the lightweight models employed in Secure-FLOATING exhibited significantly reduced FLOPs compared to their more complex counterparts. This reduction in computational complexity translates to improved efficiency and reduced resource consumption, which is a crucial consideration for real-time trajectory prediction in resource-constrained CAVs where E-CAVs are emerging as clear winners in this domain with limited battery constraints.

To further investigate the resource utilization, we assessed the CPU utilization in Table 2 during the prediction process. Remarkably, the lightweight models demonstrated efficient utilization of CPU resources, resulting in lower computational demands. This implies that the models are well-suited for deployment on hardware with limited processing capabilities. The ability to achieve accurate trajectory prediction while utilizing fewer CPU resources highlights the practicality and scalability of our approach.

Table 3 compares the training time (in seconds) for VFL models against the same models running on the nodes without the SMPC. We clearly see a reduced training time for the SMPC-based VFL, scaling up to 100 nodes. This indicate Secure-FLOATING effectiveness in performing real-time predictions and endorsements of trajectory data. Furthermore, we analyzed the MAE with respect to the increase in network size to 100 nodes for Secure-FLOATING, in comparison to individual nodes running models without federated learning shown in Table 4. Although MAE is low for small network sizes, but the effectiveness of SMPC-based VFL is clearly shown for a network size of 100 nodes predicting and endorsing trajectories of neighboring nodes. This proves the scalability of the proposed Secure-FLOATING platform in accommodating large number of nodes.

In Figure 1a, we analyze the overhead in terms of the number of bytes to better understand the scalability for message exchanges (communication rounds). We consider two sizes (1kb, and 100kb) for the message comprising the trajectories as well as the shares of the model updates, with the frequency of every 1 second, 10 seconds and 1 minute. We clearly see that using small message size has negligible overhead, while the overhead for the message size of 100kb varies, and exchanges messages every second yields the worst performance. Therefore, we recommend less frequent exchanges between nodes for both the trajectory and the model updates divided among nodes to ensure lower complexity and higher scalability. Finally, Figure 1b show the robustness of Secure-FLOATING with increase in perturbation

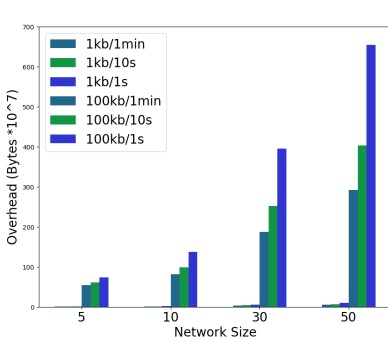

(a) Overhead comparison

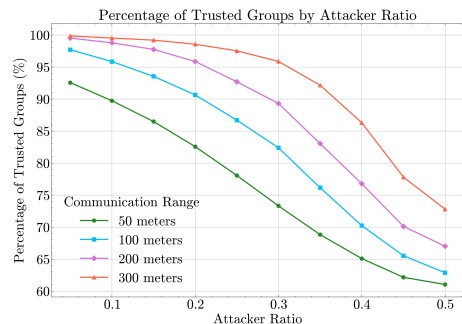

(b) Robustness against attacker comparison

Figure 1: Overhead and Robustness comparison

of node trajectories as attacker ratio vs. successfully flagging of nodes as malicious, compared with different wireless communication range defining a neighborhood. We observe, as the ratio of nodes sharing compromised trajectories increases, the nodes successfully endorsed as "Trusted" decreased. The performance of Secure-FLOATING evaluated for different communication ranges also proves its robustness, where despite the nodes being far from each other (up to 300m) and a higher attacker ratio of 0.5, we are able to successfully endorse around 75% of the nodes in the network.

Results show reduced computational and communication overhead to make Secure-FLOATING work in real-time applications in resource-constrained environments such as CAVs. We leveraged SMPC function that is simple to compute for nodes in a timely manner, where addition-based functions are relatively less complex than multiplication based or other functions. Also, the idea of employing lightweight machine learning models for predictions in the validation process is proven to be effective, where our results have shown the effectiveness of simple RNNs and LSTM based models to achieve acceptable results compared to relatively complex transformer based models on time series data.

## 6 Discussion

### Ethical Considerations

While Secure-FLOATING aims to enhance privacy and security, it is essential to consider the broader societal implications and potential risks associated with this technology. Decentralized trust models, like any machine learning system, can inadvertently perpetuate or amplify existing biases in the data. This could lead to discriminatory outcomes, where certain vehicles or groups of vehicles are unfairly penalized or favored based on factors like location, manufacturer, or driving patterns. To mitigate this, it is crucial to carefully design the trust model's learning algorithms, incorporate fairness constraints, and regularly audit the model for biases. In the event of an accident or malfunction, determining liability and responsibility in a decentralized trust system can be complex. It is crucial to establish clear guidelines and legal frameworks that address these issues, balancing the autonomy of individual nodes with the overall safety and accountability of the network. While our framework aims to enhance security through VFL and SMPC, it is essential to acknowledge that no system is completely immune to attacks. Continuous monitoring, vulnerability assessment, and the development of robust defense mechanisms are necessary to maintain the security and safety of the trust model.

### Limitations and Future Work

While Secure-FLOATING show promising results, several limitations and potential avenues for future research warrant consideration. While we have demonstrated linear scalability, the

communication overhead of the federated learning process itself could still be a bottleneck in very large-scale networks. Further research is needed to explore more communication-efficient VFL algorithms or techniques for reducing the frequency of model updates. The heterogeneity of data across different vehicles (e.g., varying sensor quality, driving patterns, and environmental conditions) can pose challenges for the convergence and generalization of the federated trust model. Techniques like personalized federated learning or model adaptation will be investigated to address this issue. While our framework incorporates several defense mechanisms against Byzantine faults and data poisoning attacks, it is essential to continuously evaluate its robustness against new and evolving attack strategies. Research into more sophisticated anomaly detection techniques and robust aggregation methods could further enhance the system's security.

Secure-FLOATING is not limited to a specific model or hardware or use-case but is designed for broader applications. CAVs research is gaining momentum and the need for such integrated solutions is important to accommodate real-time trust establishment on CAVs at scale, validate the fidelity of new multi-modal sensors (LiDARs, high-res cameras), V2X communication technologies (C-V2X, mmWave, THz), models (beyond transformers), and privacy-preserving solutions towards ensuring safety of road users. Multiple hospitals can collaborate and train models in a federated learning manner for disease diagnosis without sharing patient data directly. VFL could ensure the integrity of model updates with privacy-awareness, while SMPC enabling secure aggregation towards a consensus among them to validate diagnoses in real-time, constantly improving accuracy on different diseases diagnosis. Similarly, pharmaceutical companies can perform clinical trials across multiple institutions for real-time data validation while preserving patient privacy and data integrity. Financial institutions can collaborate and train models for fraud detection without sharing sensitive customer transaction data. VFL and SMPC could ensure privacy and security, while consensus could be used to validate transactions in real-time, towards faster and more accurate fraud detection.

The current trust model is primarily focused on accuracy and robustness. However, it is crucial to develop SMPC-based trust models that are also interpretable and explainable. This would allow human operators or regulators to understand the reasoning behind the trust scores and intervene if necessary. The design and implementation of effective incentives that encourage honest participation and deter malicious behavior remain an open research problem. We are actively exploring solutions that leverage reinforcement Q-learning within a federated learning framework to design effective incentive schemes and personalize the system's behavior based on individual node constraints and data characteristics. Deploying Secure-FLOATING in a real-world network would involve additional challenges, such as dealing with intermittent connectivity, varying network topologies, and legal and regulatory considerations. These aspects would need to be carefully addressed in future research.

In addition to the above-mentioned research directions, we will explore more efficient VFL algorithms and investigate incentive mechanisms to encourage wider participation in the decentralized trust network. Ultimately, our goal is to pave the way for the widespread adoption of CAVs, ensuring that they operate in a secure, private, and cooperative manner.

## 7    CONCLUSIONS

We proposed Secure-FLOATING, a novel framework for decentralized trust establishment in real-time for CAVs and other nodes navigating around each other. We propose (1) a tailored VFL algorithm that preserves privacy while ensuring the integrity of model updates through zero-knowledge proofs, (2) SMPC protocol for secure model aggregation with rigorous privacy guarantees, and (3) a scalable and secure blockchain based consensus mechanism for trajectory validation with provable linear complexity making it not only theoretically sound, as demonstrated by formal proofs for privacy, security, and scalability, but also practically effective, as evidenced by our extensive evaluation based on real-world data from New York City. Results show that Secure-FLOATING is an efficient trust model in terms of accuracy, privacy preservation, robustness, and scalability and represents a significant step towards the development of trustworthy and reliable autonomous transportation systems.

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

# A  Full Proofs

## A.1  Proof of Theorem 4.1

Notations. (n): Number of participating nodes, $(\theta_{v_i} \in \mathbb{R}^d)$: Local model update (gradient vector) of node $(v_i)$, $(\theta_{global} \in \mathbb{R}^d)$: Global model update (aggregated gradient vector): $(\theta_{global} = \sum_{v_i=1}^n \theta_{v_i})$, $(\Delta\theta_{global})$: The $(\ell_2)$-sensitivity of the aggregation function: $(\Delta\theta_{global} = \max_{v_i} |\theta_{v_i}|_2)$, $(Lap(\lambda))$: Laplace distribution with scale parameter $(\lambda)$, $(M(D))$: The output of the MPC aggregation protocol on dataset (D), $(D, D')$: Neighboring datasets differing in a single node's trajectory data.

**Sensitivity Analysis:**  The aggregation function is simply the sum of model updates. The sensitivity, $(\Delta\theta_{global})$, is the maximum change in the output (aggregated update) caused by changing a single node's input (local update). Since we are using the $\ell_2$ norm to measure the distance between updates, the sensitivity is the maximum $\ell_2$ norm of any individual update:

$$\Delta\theta_{global} = \max_{v_i} \|\theta_{v_i}\|_2$$

**Noise Addition and Aggregation:**  In our SMPC protocol, each node $v_i$ adds Laplacian noise with scale $(\lambda = \Delta x/\epsilon)$ to its model update before secret sharing:

$$\tilde{\theta}_{v_i} = \theta_{v_i} + Lap(\Delta\theta_{global}/\epsilon)$$

The noisy model updates are then aggregated on the distributed ledger as:

$$\tilde{\theta}_{global} = \sum_{v_i=1}^n \tilde{\theta}_{v_i}$$

**Privacy Analysis:** Let $(S \subseteq \mathbb{R}^d)$ be any subset of possible outputs. Consider $(M(D))$ as the output of the MPC aggregation protocol on dataset (D), we want show that:

$$Pr[M(D) \in S] \le e^\epsilon Pr[M(D') \in S] + \delta$$

We start by analyzing the probability density functions (PDFs) of the Laplace distribution. For any $(z \in \mathbb{R}^d)$, the PDF of a Laplace distribution with scale parameter $(\lambda)$ is:

$$pdf_{Lap(\lambda)}(z) = \frac{1}{2\lambda}\exp(-\|z\|_1/\lambda)$$

where $(|z|_1)$ is the $(\ell_1)$ norm of $(z)$.

Now, consider the probability density of the noisy aggregate $(\tilde{\theta}_{global})$ given dataset $(D)$. Since the noise added to each $(\theta_{v_i})$ is independent, we have:

$$Pr[M(D) = \tilde{\theta}_{global}] = \prod_{v_i=1}^n pdf_{Lap(\lambda)}(\tilde{\theta}_{v_i} - \theta_{v_i})$$

Similarly, for the neighboring dataset (D'), which differs from (D) in only one node's data (say, node k), we have:

$$Pr[M(D') = \tilde{\theta}_{global}] = pdf_{Lap(\lambda)}(\tilde{\theta}_{v_k} - \theta'_{v_k}) \prod_{i \ne k} pdf_{Lap(\lambda)}(\tilde{\theta}_{v_i} - \theta_{v_i}))$$

Taking the ratio of these probabilities, we get:

$$\frac{Pr[M(D) = \tilde{\theta}_{global}]}{Pr[M(D') = \tilde{\theta}_{global}]} = \frac{pdf_{Lap(\lambda)}(\tilde{\theta}_{v_k} - \theta_{v_k})}{pdf_{Lap(\lambda)}(\tilde{\theta}_{v_k} - \theta'_{v_k})}$$

**Applying Laplace PDF Properties:** Substituting the Laplace PDF, we get:

$$\frac{Pr[M(D) = \tilde{\theta}_{global}]}{Pr[M(D') = \tilde{\theta}_{global}]} = \exp\left(\frac{\|\tilde{\theta}_{v_k} - \theta'_{v_k}\|_1 - \|\tilde{\theta}_{v_k} - \theta_{v_k}\|_1}{\lambda}\right)$$

Using the triangle inequality, we can bound the numerator:

$$\|\tilde{\theta}_{v_k} - \theta'_{v_k}\|_1 - \|\tilde{\theta}_{v_k} - \theta_{v_k}\|_1 \leq \|\theta_{v_k} - \theta'_{v_k}\|_1 \leq \sqrt{d}\|\theta_{v_k} - \theta'_{v_k}\|_2 \leq \sqrt{d}\Delta\theta_{global}$$

since $(|\cdot|_1 \leq \sqrt{d}|\cdot|_2)$ for any vector in $(\mathbb{R}^d)$.

Therefore:

$$\frac{Pr[M(D) = \tilde{\theta}_{global}]}{Pr[M(D') = \tilde{\theta}_{global}]} \leq \exp(\sqrt{d}\Delta\theta_{global}/\lambda)$$

**Setting Noise Level:** To ensure $(\epsilon, 0)$-differential privacy, we need this ratio to be at most $(e^\epsilon)$. Thus, we set the noise level as:

$$\lambda = \frac{\sqrt{d}\Delta\theta_{global}}{\epsilon}$$

**Accounting for $\delta$:** Since the Laplace distribution has exponential tails, there's a small probability $\delta$ that the noise doesn't sufficiently mask the difference between datasets. To account for this, we use the advanced composition theorem of differential privacy, which tells us that after T rounds of aggregation, the overall privacy guarantee becomes $(\epsilon', \delta')$, where:

$\epsilon' = \sqrt{2T\ln(1/\delta)}\epsilon + T\epsilon(e^\epsilon - 1)$, $\delta' = T\delta$ We can choose $\epsilon$ and $\delta$ for each round appropriately to guarantee the desired $(\epsilon', \delta')$ privacy for the entire process.

Therefore, we conclude that the MPC aggregation protocol satisfies $(\epsilon, \delta)$-differential privacy.

