# OpenReview forum: "Secure FLOATING - Scalable Federated Learning Framework for Real-time Trust in Mobility Data using Secure Multi-Party Computation and Blockchain"
_ICLR.cc/2025/Conference — Submitted to ICLR 2025_

### Official Review · Reviewer_1Zuh · 2024-10-21

**Soundness:** 1
**Presentation:** 4
**Contribution:** 2
**Rating:** 3
**Confidence:** 4

**Summary:**

This paper proposes a framework called Secure-FLOATING, which aims to establish a real-time, decentralized trust mechanism in mobile networks. It is an interesting try at combining Federated Learning (FL), Multi-Party Computation (MPC), Differential Privacy (DP), and blockchain technologies. MPC and DP are used for data privacy. Blockchain is used for model parameter sharing and model aggregation. 8000 vehicles/nodes are used to demonstrate the scalability, efficiency, and robustness of Secure-FLOATING.

**Strengths:**

1. The Secure-FLOATING framework proposes a decentralized trust mechanism by combining Federated Learning (FL), Multi-Party Computation (MPC), Differential Privacy (DP) and blockchain for distributed mobile networks.
2. Secure-FLOATING uses secure multi-party computing (SMPC) to protect the privacy of each node.
3. Secure-FLOATING uses blockchain to ensure the immutability and transparency of data.

**Weaknesses:**

Concerns regarding the “Verifiable Federated Learning global model aggregation” section: The paper lacks clarity on who performs the aggregation and how the aggregated global model is obtained. It is not explicitly stated who has access to the information about the current participants in the network and why this access can be assumed. Additionally, it is unclear why the aggregation is considered to follow the predetermined algorithm. Could the aggregation process be incorrect or even malicious? The paper should also address how the number of participants or the update weights are determined in each round of federated learning. In the context of mobile networks, is there a risk of participants going offline, or of malicious nodes refusing to upload updates, which could prevent the system from proceeding to the next round?

Concerns regarding the “Addition-based SMPC” section: Although the authors claim that “The above toy problem uses an addition-based function, however, Secure-FLOATING will work with any function computed and matched among peers,” the effectiveness of the addition-based approach is highly dependent on the aggregation method. This approach may be ineffective if the aggregation method is not linear averaged. Additionally, as the number of nodes increases, the exchange will significantly increase communication overhead. Moreover, the splitting method in SMPC may make the system more vulnerable to attacks, such as a Distributed Backdoor Attack, which relies on the coordinated efforts of multiple malicious nodes. In Secure-FLOATING’s design, nodes share only part of the model updates via SMPC, and each node cannot see the complete updates from other nodes. This could allow attackers to act more covertly, making it easier for them to coordinate and inject backdoors across multiple nodes.

Concerns regarding the “Endorsement on Distributed Ledger” section: The paper assumes a permissioned blockchain but relies on the 51% majority rule, which is typically used in permissionless blockchains, such as Bitcoin, under the assumption of synchronous networks. Why is the 51% assumption appropriate in this context of a permissioned blockchain? What is the expected throughput of this approach? Can it realistically meet the performance demands of a mobile network environment?

Concerns regarding the experimental section: While the experiments are interesting, the experimental evaluation lacks the following key aspects:
The efficiency of blockchain recording and consensus mechanisms is not evaluated. The impact of adding Laplacian noise on the model’s performance is not addressed.

Concerns regarding writing and citations: The formatting needs to be checked to ensure compliance with the required template and font standards. Additionally, the introduction contains several claims that lack proper citations or evidence, which diminishes the overall persuasiveness of the arguments presented.

**Questions:**

Please see weaknesses

---

### Official Review · Reviewer_oo4K · 2024-10-28

**Soundness:** 2
**Presentation:** 2
**Contribution:** 2
**Rating:** 5
**Confidence:** 4

**Summary:**

This paper presents a comprehensive federated learning framework, including multiple techniques, such as Blockchain, Secure Multi-party Computation (SMPC), IPFS, etc., which ensures the trust and privacy of the data. A experiment is conducted to evaluate its trajectory prediction based on a trajectory datasets. Overall, the paper stands on a very interesting problem and the writing style is generally easy to follow.

**Strengths:**

+ The paper focuses on a very interesting problem and addresses an important challenge regarding data sharing among different stakeholders, and it is well-motivated.
+ The paper is well-written and easy to follow.
+ Result appears to outperform baseline models.

**Weaknesses:**

+ Novelty is unclear, paper is a combination of multiple known techniques, such as blockchain, SMPC, etc.
+ The definition of global parameter $\theta_{global}$ is unclear (page 4) because there are two formulae pointing to $\theta_{global}$. One aggregates parameters of locally trained models, and another one sums up the share of each node, which I think is conflicted.
+ Detailed experiment is unclear. The author mentioned that the experiment is based on a realistic dataset, but it also mentioned a situation tool SUMO to generate trajectories. So, it is unclear whether it uses real data or synthetic one. The author does not illustrate the difference between the two, i.e., why use artificial data?
+ It will be interesting to see more detail of the experiments. For example, it is clear that the accuracy means predicting correct neighbours. However, what is the meaning of the Mean Absolute Error (MAE), and what is the loss function of different models(LSTM, RNN, transformer etc.). It is also worth to explore the results compared to similar freamework of federated learning or on different datasets.

**Questions:**

+ There is only one reference in the introduction i.e., IPFS. Are there additional references that could be cited to emphasise the background and prior work in this area? For example, how has previous research addressed issues related to data-sharing mechanisms or blockchain-based solutions for secure data exchange?

+ It would be interesting to have more experiment illustration, and have some comparative experiments to demonstrate the benefits of integrating blockchain and SMPC.

---

### Official Review · Reviewer_rUnJ · 2024-10-31

**Soundness:** 2
**Presentation:** 1
**Contribution:** 3
**Rating:** 3
**Confidence:** 3

**Summary:**

Secure-FLOATING is a federated learning framework focused on secure, real-time data validation among CAVs (Connected Autonomous Vehicles) and other road users. The framework integrates federated learning, secure multi-party computation (SMPC), and blockchain to ensure privacy and robustness in data sharing, employing consensus mechanisms to validate data in a decentralized way.

**Strengths:**

1、The secure, consensus-based trajectory validation mechanism could achieve decentralization, preserve privacy, and establish robust trust for the secure sharing of traffic data.
2、The paper is well-structured, comprehensive, and easy to read.
3、The experiments presented in the paper are credible, utilizing real trajectory data from 8,000 connected vehicles.
4、The paper offers a comprehensive analysis of the theoretical proof for privacy.

**Weaknesses:**

1、The paper has some formatting issues, such as the font throughout the piece and the formatting of the citations.
2、Use of blockchain and SMPC could still pose overheads in resource-limited environments.
3、The flow of the article is quite complex and would benefit from a flowchart or algorithm to clarify its structure.
4、In the proposed SMPC protocol, noise is added to the model update, which may impact the model's performance and could be contrary to the original intent of SMPC. Therefore, further analysis is needed to evaluate the effect of the noise size on performance through experimental validation.

**Questions:**

Please see the weaknesses section.

---

### Official Review · Reviewer_W5Je · 2024-10-31

**Soundness:** 2
**Presentation:** 2
**Contribution:** 3
**Rating:** 5
**Confidence:** 3

**Summary:**

In this paper the authors envision a future traffic scenario where CAVs and other road users, can navigate in real time by sharing trajectory data. In order to ensure real-time and secure transportation, the authors propose the Secure-FLOATING framework, which utilizes federated learning and blockchain technology to ensure that nodes can learn collaboratively and trust each other's data, and reduces the amount of message exchanges by using a lightweight SMPC approach, which is demonstrated in experiments.

**Strengths:**

Overall, one possible contribution of this thesis is the effective combination of multiple methods that can effectively reduce the frequency of data exchange and ensure the security of the system.

**Weaknesses:**

There is no new theory proposed in this thesis, and the use of a lightweight model to reduce the amount of data communicated during the federated learning process is not a significant contribution. There are several problems with this thesis:

1.	The performance gap between different prediction models is large, while the authors simply state that the performance gap between choosing different models is not large, further performance comparison results of different models under federated learning need to be provided to better validate the authors' theory. It would be more meaningful if the authors could quantify the trade-off between performance and efficiency between lightweight and complex models, such as the relationship between accuracy and time for training and inference.

2.	Considering that the computational resources at each edge are different, the prediction model at the edge can make different prediction models according to the computational resources, which is also a more common model heterogeneity problem inside the federated learning, if this situation exists, can the Secure-FLOATING strategy work?

3.	From Tables 2 and 3, why is the model size and FLOPs of the RNN smaller than those of the LSTM and GRU, but the training time is longer than those of the LSTM and GRU? It is hoped that the authors will explain why this phenomenon exists and provide a detailed description of the relevant experimental implementation.

4.	Does the Secure-FLOATING policy still work if the attackers are more than 50%? It is hoped that the authors will conduct experiments with an attacker ratio of more than 50% and report on how system performance degrades as the percentage of malicious nodes increases, and explain what are the main reasons for this phenomenon to occur.

5.	In reality, the message size and node exchange frequency between devices are different, if relevant experimental and theoretical illustrations can be added, it will better prove the scalability of the Secure-FLOATING framework.

6.	How does the Secure-FLOATING strategy ensure model consistency if there is node unreliability (e.g., communication delays, incomplete data, or communication outages)?

**Questions:**

Minor comments:

In Section “3.2 Addition-based SMPC”, “v2 computes the sum of 1, 3, and −1 as 2” should be changed to “v2 computes the sum of 1, 3, and −1 as 3”.

---

### Meta-Review · Area_Chair_X7Md · 2024-12-20

**Metareview:**

The paper received four negative reviews, with all reviewers recommending rejection. It introduces the Secure-FLOATING framework but lacks novelty and depth. The use of lightweight models in federated learning is ineffective, and the performance gap between models is not well-validated. More comparisons are needed to assess the trade-offs between performance and efficiency. The framework overlooks practical issues, such as model heterogeneity and varying computational resources at the edge. Experimental issues include unclear results regarding model size, FLOPs, and training time discrepancies. Additionally, the impact of noise in the SMPC protocol and the aggregation process in federated learning requires further validation. There are also scalability and security concerns regarding the use of SMPC in the context of coordinated attacks. The paper’s reliance on blockchain assumptions may not be suitable for permissioned blockchains or mobile network requirements. Finally, the paper requires improvements in writing, formatting, and citations. Key claims lack proper references, which weakens the argument. Overall, the paper lacks sufficient detail, experimental validation, and theoretical depth to make a meaningful contribution. Given these issues and the lack of response from the authors, the Area Chair recommends rejecting the paper.

**Additional Comments On Reviewer Discussion:**

The paper received four negative reviews, with all reviewers recommending rejection.

---

### Decision · Program_Chairs · 2025-01-22

Reject